# On the Use of a Rotatable ECT Sensor to Investigate Dense Phase Flow: A Feasibility Study

**DOI:** 10.3390/s20174854

**Published:** 2020-08-27

**Authors:** Radosław Wajman, Robert Banasiak, Laurent Babout

**Affiliations:** Institute of Applied Computer Science at Lodz University of Technology, Stefanowskiego 18/22, 90-924 Lodz, Poland; robert.banasiak@p.lodz.pl (R.B.); laurent.babout@p.lodz.pl (L.B.)

**Keywords:** pneumatic conveying, electrical capacitance tomography, image reconstruction, sensor

## Abstract

This paper presents the feasibility study of dynamic flow measurements using the concept of a rotatable electrical capacitance tomography (ECT) sensor. The experiment considered horizontal flow in a pneumatic conveying flow loop in the case of dense phase flow. Slugs and settled layers were imaged and a comparison was made between no rotation or rotation of the sensor for two image reconstruction schemas: linear back projection (LBP) and non-linear iterative back projection. Data were evaluated both qualitatively and quantitatively by estimating the solids concentration level for different hue levels.

## 1. Introduction

Electrical capacitance tomography (ECT) has proven to be a very suitable technique for flow control applications such as flow rate, flow regime, and real-time material distribution and concentration estimation [1,2,3,4,5]. One of the standard ways is to reconstruct images based on the different capacitance measurements obtained between the different pairs of electrodes. Image quality and accuracy are key issues to obtain reliable results, and this is still a challenging problem, even after almost 30 years of research in the field, mainly because of two main drawbacks: the non-linearity of the electric field and ill-posed problems. Considerable work has been done on the image reconstruction side to enhance accuracy [6,7]. In the same way, acquisition units have been continuously improved to achieve a higher signal-to-noise ratio (SNR) and measurement precision [8]. Sensor optimization has also been studied. In that respect, the increased number of electrodes in the sensor is one key issue to reduce the ill-posedness that the technique faces. However, the increase is limited by the spatial constraint of the sensing area. Increasing the number of capacitance electrodes while keeping the capacitance range into a measurable range implies that the shorter electrode dimension in the radial direction is compensated by increasing their length (in the axial direction). This leads to a larger averaging zone, which in most applications, is not acceptable. Increasing the number of electrodes also implies a more complicated and expensive acquisition unit, which somehow goes against one of the main advantages of the technique: its relatively low price. Therefore, new strategies have been thought of to increase the number of measurements.

The two main proposed solutions consider the rotation of the electric field, either electrically or mechanically. Olmos et al. [9] proposed to create a four capacitance electrodes sensor by linking *N* segments in four groups, treated as single. Their method, by adding the different results obtained for the different segment combinations, increased the sensitivity of the measured capacitances and the image accuracy. The authors also claimed that the combination of 20 segments grouped into four electrodes with five possible rotations achieves higher inter-electrode capacitances and broader sensing areas comparing to a conventional 12-electrode scheme. However, the authors did not describe the advantage of using this configuration instead of a classical sensor with 20 electrodes. Mechanical movement of the electrodes can also enhance image accuracy, despite reducing the fast read-out capability of the system because of the sensor rotation. The method has been adopted for electrical impedance tomography with a rotatable sensor controlled by a stepper motor [10]. Murphy and York [11] proposed to mount electrodes on the impeller in a mixing vessel to increase the number of measurements obtained between these rotating electrodes and the fixed ones on the vessel inner wall. In the case of capacitance tomography, a similar method was proposed earlier by Frounchi and Bazzazi [12], but the sensor only considered four electrodes and simulation results. Later, Liu et al. [13] extended the approach to account for 16 electrodes. Their results showed, that this electrode configuration with the shortest rotation time for three intermediate steps, which is used to cover the angle between two adjacent electrodes, results in the highest image accuracy.

The present paper aims to extend the study carried out by Liu et al. on static objects to a more realistic application that considers dynamic flow. Pneumatic conveying was chosen for this study as it has been widely studied using ECT [14,15,16,17,18,19,20]. The prototype capacitance sensor was modified to be mounted on the pneumatic conveying flow loop of the Tom Dyakowski Process Tomography Laboratory at the Lodz University of Technology. Measurements were first conducted on the horizontal section of the rig in which polyamide granulates are pneumatically transported. The usefulness of the rotatable ECT sensor was studied as a function of the solids flow rate to provide more accurate information about the distribution and the concentration of media. The latter is obtained using two planes of eight electrodes surrounding the rotatable sensor. This paper is organized as follows. Section 2 focuses on the experimental set-up. This section is followed by a description of the image reconstruction strategy before results for the dynamic experiment are presented and discussed in Section 4.

## 2. Experimental Setup

### 2.1. The Pneumatic Conveying Flow Loop

Dynamic tests were performed on the solid/gas pneumatic flow rig at the Tom Dyakowski Process Tomography Laboratory. A schematic description of the installation is shown in Figure 1.

The installation uses two tanks: one to feed the installation with solids and the second one to collect them. The solids are injected in the loop using a rotary feeder that can operate at a frequency of up to 120 Hz. At the current state, the tank contains 120 kg of solids. The control of the emptying/refilling process is performed using weight cells supporting the collecting tank. The cells are also used to estimate the mass flow rate. It is done with the standard procedure that corresponds to the weight increase calculation in the top tank while the gate valve below the bottom tank has been closed. Air from the compressor is cooled and sent to the rotary feeder, where it entrains the particles. The air injection is monitored with a frequency up to 50 Hz. The installation of sonic nozzles and a pressure transducer allows the estimation of the gas flow rate. Different flow regimes can be obtained by changing the corresponding frequencies. It was shown that slug flow can be observed at 15 Hz and 75 Hz for the solids and air injections, respectively. This corresponds to a solid mass flow rate of ~1.4 t/h and an estimated gas flow rate in an empty pipe of 2–3 m/s. Concerning the solids flow rate, it is calculated using capacitance measurements, as described in the next sub-section. The two horizontal sections are about 10 m long while the vertical part is 6 m long. The inner pipe diameter is 65 mm and in the horizontal and vertical sensing areas which correspond to Perspex tubes of 1.2 m long, the wall thickness is 2 mm. The solid particles are made of polyamide and have a typical size of 1–3 mm and a density of 1020 kg m^−3^. In the present paper, only sensors were mounted on the horizontal sensing area, as described in the next sub-section.

### 2.2. ECT Sensor and Acquisition Hardware

Two types of capacitance sensors were used for this study, as it is shown in Figure 2.

A dual-plane sensor made of eight electrodes was used for solids flow velocity calculation. The electrodes were connected to the 16 AC-base ECT system. The interspacing was set to 50 cm so that it left enough space to mount in between the rotary sensor and to provide enough distance for the velocity measurement of the solids. Both sensors were calibrated using the same sample for the full condition, which corresponds to a 65 mm diameter polyamide cylinder of 3.2 relative permittivity. Only the electrode holder of the rotary ECT sensor was changed to fit the flow rig. Indeed, an I-shape has been specially designed in order to match the size of the frame (typically designed to hold a 150 mm pipe) and the outer diameter of the flow rig. This can be seen in Figure 3.

Another modification considered the installation of discs at the ends of the frame in order to fix the sensor on the flow facility. Twelve electrodes were also mounted on the rotatable sensor on the inner wall of the holder. The electrode size was 90 × 15 mm.

The rotatable sensor was connected to an electrical process tomography system that was developed at the Lodz University of Technology [21] in a frame of DENIDIA project (referred hereafter as DECaRT — Denidia Electrical Capacitance and Resistance Tomography). The system comprises 32 channels that work in AC mode and a special communication software interface was designed to control the stepper motor rotation synchronously with the capacitance measurements, as it is shown in Figure 2.

During flow measurement, it is critical to minimize the rotation time between two successive sets of capacitance measurements (in the present case, a set is composed of 66 capacitance measurements). Even if the motor can have an angular speed of up to 150 rpm, it is a value that cannot be achieved in our present case because of the significant vibrations that are generated during the rotation. Moreover, problems can also occur from the set of gear that has, in the present design, difficulty coping with a precise rotation step at such a speed. In the present state, the rotary sensor works well, in terms of rotation accuracy and low generated vibration for an angular speed up to 25 rpm. In the case of 12 electrodes and three rotation steps, this corresponds to a minimum rotation time of about 100 ms. In that context, the cardinality of the set of combined measurements is equivalent to the one that would be obtained for a sensor equipped with one plane of 24 electrodes. Since the acquisition system can record a single capacitance measurement in 0.1 ms, the acquisition speed with the above configuration is about 3–4 frames/s. It is lower than the frame rate provided by the dual-plane system, i.e., around 55 frames/s. The effect of this difference will be discussed in Section 4.2. In such a context, a clockwise–anticlockwise rotation strategy was adopted during acquisition. Firstly, a set of capacitances is measured. Then, three successive clockwise rotations of 7.5° followed by capacitance measurements are performed (the rotation step is calculated by dividing 360° by the number of electrodes and the number of rotation steps +1). When the last position is reached and the acquisition is made, the I-shape tube rotates anticlockwise another three times, the initial set corresponding to the last one acquired during the clockwise rotation. In such a context, measurements 1, …, 4 in one rotation direction correspond to measurements 4, …, 1 in the other rotation direction.

## 3. Image Reconstruction

The main goal of an inverse problem solution is to approximate the permittivity distribution inside a sensor using simulated or experimental capacitance data and accurate electric field sensitivity analysis. The sensitivity analysis in ECT imaging is typically based on equations expressing the energy of the electric field accumulated in the sensor space [22,23]. The standard methods that use the finite element method (FEM) can calculate the electric field at each element (for instance, with triangle shape) of the FEM mesh. Banasiak et al. [24,25] showed that the sensitivity matrix can also be calculated at nodes. It helps to reduce the number of values consequently in the sensitivity matrix and can decrease the under-determination degree of an inverse problem. Generally, the equation for nodal-based sensitivity matrix can be expressed as:(1)$$S_{j}=\frac{1}{U^{2}}\int_{\Omega }N_{j}\left(x,y\right)\vec{E^{2}}\left(x,y\right)d\Omega$$

It can be observed that the size of the nodal sensitivity matrix is approximately 50% smaller than traditional sensitivity matrices calculated for the mesh elements. This is an improvement that can help to reduce the image reconstruction computational effort.

In this paper, two deterministic reconstruction techniques were applied in a rotatable scheme: simple and fast linear back projection (LBP) and slow but accurate non-linear iterative back projection (IBP) according to a solution proposed by Banasiak at al. [26]. The methodology of the generation of a global rotation-based sensitivity matrix requires one to calculate a set of sensitivity maps for all the rotation positions and differs from other classical approaches [27,28]. The rotation of the electrodes around the media is simulated using the finite element method (FEM) calculation by changing the set of nodes that define each electrode.

The solution to an image reconstruction problem [27,29] can be a function g that resolves an image of electrical permittivity distribution $\hat{\varepsilon }\left(x,y\right)$ inside a sensor from experimental or simulated capacitance data ***C_m_*** according to this formula:(2)$$g:C\mapsto \hat{\varepsilon },$$

In the next step, we must solve:(3)$$\hat{\varepsilon }=S^{+}\cdot C_{m},$$
where ***S^+^*** is a sensitivity (Jacobian) matrix. This task, commonly named linear back projection (LBP), is especially difficult because matrix ***S*** is not rectangular. Additionally, the set of Equation (3) is under-determined.

LBP was one of the first algorithms used to reconstruct an image from capacitance data and this is a single-step image reconstruction algorithm that is commonly used in ECT systems with non-rotatable circular-shaped sensors now. For the rotatable ECT sensor, it takes the same format as for the non-rotatable circular-shaped sensors with a rotation-based sensitivity matrix. The version of this technique for LBP can be defined as follows in the case of 12 electrodes and four positions during sensor rotation:(4)$$\hat{\varepsilon }\left(x,y\right)=f_{TH}\left[\frac{\sum_{i=1\;}^{11}\sum_{j=i+1}^{12}\;C_{{\left(p=1,2,3,4\right)}_{ij}}\;S_{\left(p=1,2,3,4\right)}^{+}\left(x,y\right)}{\sum_{i=1}^{11}\sum_{j=i+1\;}^{12}S_{\left(p=1,2,3,4\right)}^{+}\left(x,y\right)}\right],$$
(5)$$f_{TH}\left(x,y\right)=\{\begin{array}{c} 0\\ \hat{\varepsilon }\left(x,y\right)\;\\ 1 \end{array}\begin{array}{c} x\leq 0\\ 0<\hat{\varepsilon }\left(x,y\right)<1\\ x\geq 1 \end{array}$$

LBP is a fast algorithm. The single image reconstruction can take about 15 ms for modern PCs. Nevertheless, it does not go hand in hand with the accuracy. To find an accurate inverse problem solution, an iterative back-projection algorithm in a fully nonlinear scheme [26] was applied to a rotational-based set of measurement data (called hereafter IBP). The IBP reconstruction process we used for the rotational scheme can be divided into the following steps:Set *k = 0*;Find an initial approximate image ${\hat{\varepsilon }}_{\left(k\right)}$ using an experimental set of capacitance data ***C_m_*** and the LBP algorithm with an initial nodal sensitivity matrix $S_{\left(k\right)}^{+}$ calculated for homogeneous media with the normalized relative permittivity value of 0.5 (Equations (4) and (5)). The value of dielectric constant for simulating homogeneous sensor forward model was estimated due to the known minimum (1.0) and maximum (3.2) relative permittivity values of air and polyamide particle, respectively;Set *k = k + 1*;Find a forward solution as a new capacitance matrix $C_{c\left(k\right)\left(p\right)}$ and electric field distribution $V_{\left(k\right)\left(p\right)}$ for each *p* positions (*p* = 1, 2, 3, 4) using ECT rotational FEM forward model and previously reconstructed image ${\hat{\varepsilon }}_{\left(k\right)}$;Normalize the new four capacitance sets $C_{c\left(k\right)\left(p\right)}$ (*p* = 1, 2, 3, 4);Combine the global rotation-based capacitance data set $C_{c\left(k\right)}$;Calculate an updated sensitivity matrix $S_{\left(k\right)\left(p\right)}^{+}$ using $V_{\left(k\right)\left(p\right)}$ and nodal algorithm for each *p* positions (*p =* 1, 2, 3, 4);Combine the global matrix $S_{\left(k\right)}^{+}$ using four rotation-based sensitivity maps;Normalize the global sensitivity matrix $S_{\left(k\right)}^{+}$;Find the approximate image ${\hat{\varepsilon }}_{\left(k+1\right)}$ using an iterative optimization function that can be typically expressed as:(6)$${\hat{\varepsilon }}_{\left(k+1\right)}={\hat{\varepsilon }}_{\left(k\right)}-\alpha \cdot S_{\left(k\right)}^{+}\left(C_{c\left(k\right)}-C_{m}\right),$$
(7)$$C_{c\left(k\right)},V_{\left(k\right)\left(p\right)}=FEM\left({\hat{\varepsilon }}_{\left(k\right)},p\right),\;p\;=\;1,\;2,\;3\;,4$$
(8)$$S_{\left(k\right)}^{+}={\left[nodal\left(V_{\left(k\right)\left(p\right)},p\right)\right]}^{T},$$Repeat steps 3–10 until *k* < *desired number of iterations*.

Each *k*-th step of the whole IBP image reconstruction process requires solving the forward problem for *V_(k)_* and $C_{c\left(k\right)}$ using the rotational scheme. In that time, the sensitivity matrix $S_{\left(k\right)}^{+}$ is also updated using current *k*-th reconstructed image and the given rotation angle. The convergence of image reconstruction can be typically adjusted by the relaxation factor *α*. In this paper, the *α* factor was chosen empirically from the value range: 0.01–0.1. There was no strict stop criterion applied in the iterative reconstruction scheme due to a lack of proper reference phantom image to compare with. The only capacitance residual error was observed to select the desired typical number of iterations for all cases according to the formula given by Yang and Peng [6]. It is also worth mentioning that the IBP algorithm is time consuming compared to LBP (~120 times slower), which makes it so far only applicable off-line [30].

## 4. Results and Discussion

### 4.1. Experimental Results

The main aim of this paper was to focus on the flow regime when a stratified or settled bed occurs. This is the case during the slug regime or stratified flow. In the case of the former one, when the gas flow rate is below the saltation velocity [15], the settled layer of solids is picked up by slugs (flow instabilities) that are the results of gas pockets, then mixed and conveyed along the pipe for some distance. Solids are dropped off back from a new settled layer with a new concentration level. In that context, the concentration and also the shape of the settled layer are interesting parameters to characterize since they relate to the second gas pocket that pushes the slug. The main drawback of using ECT is in the shape/concentration recovery of an object that is significantly prone to smoothing/overestimation due to the ill-posedness of the reconstruction. To state at which extent these parameters can be obtained with a better accuracy using the rotatable sensor, real-time study of the above-mentioned slug flow regime was carried out. It was performed with a particular focus on both the settled layer and slug (even the last one may present less interest since slugs usually fill the tube space). While the first study focused on the potential to estimate the settled layer concentration better, the second one was rather devoted to determining the slug concentration. This, as it is discussed later on in this section, is directly dependent on the slug speed and size. To complete the study, we also compared the results for two reconstruction strategies mentioned in the previous section, and two cases: with or without rotation. The latter comparison uses the same set of data since no rotation corresponds to the first position in each rotation sequence.

Figure 4 shows an exemplary case of the slug regime that is analyzed in this section and recorded from the first plane of the dual-plane ECT system. The beginning of the solids injection is characterized by a concentration build-up that is mainly caused by a stratified flow regime until stability occurs (after around 20 s), before a first slug, for which the average concentration profile is magnified in Figure 4, is propelled. During the remaining 90 s, a succession of slugs and settled layers with different levels of concentration are recorded. From this set of events, five were retained for further analysis, among which three corresponded to slugs (3, 4, 5) and two to settled layers of different concentrations (1, 2). These particular events were chosen for their quasi-steady-state nature. This also assumes that a slug density does not vary significantly along its length. In that manner, these events can be considered as static objects of different kinds (density levels for slug, heights for layers).

The corresponding reconstructed images for each event are presented in Figure 5.

The figure presents the results obtained for both image reconstruction methods, i.e., LBP and IBP and taking into account 1 (i.e., no rotation) or 4 positions (i.e., rotation), together with a color scale that corresponds to the normalized permittivity values. In the case of IBP, a relaxation factor of 0.1 and 20 iterations were used for each presented case. The figure also shows the corresponding view, from the video camera, that surrounds the rotatable sensor and gives an estimate of the solids level in the pipe. The rotatable sensor covers the transparent pipe. Therefore, the red rectangles are used to highlight the regions where the flow changes may be observed. It is worth mentioning for clarity that some of the normalized capacitances, even after sensor calibration, did present values much larger than 1 in the case of the rotatable sensor. Consequently, the corresponding reconstructed image did present normalized relative permittivity values much larger than 1 that were rescaled to 1, which explains the presence of red patches at the vicinity of adjacent electrodes where the capacitance is the highest. This is a problem that is frequently met when the capacitance measurements are not stable enough when the SNR is too low. The reasons for such saturation are given in the next section. The plan for future research is to implement for the LBP algorithm the division operation to reduce these artifacts in the near-wall regions as it was developed in reference [31]. One can see that the images from IBP present sharper boundaries of the solids body than for LBP, especially for the settled layers #1 and #2. The noise that is present in the middle of the sensing area is also filtered. The improvement from the rotation is, however, not clearly visible, despite the appearance of the layer edge in the case of LBP with rotation for the settled layer #1. Additionally, the gap that is present between regions of high permittivity (red) and low permittivity (cyan) is reduced for IBP with rotation (compare 1st row, 4th and 5th columns in Figure 5). Concerning the slugs (#3 to #5), differences can be noted between no rotation and rotation in the case of LBP. However, different cases correspond to different acquisition situations. In the case of slug #3, a decreasing gradient of normalized relative permittivity distribution from the bottom to the top of the image appears in the case of LBP/rotation. At the same time, it is invisible when image reconstruction does not take into account rotation. This gradient is not present in the case of slug #4, for both the rotation and no rotation cases, despite a more uniform relative permittivity distribution in the case of no rotation. The images related to slugs #3 and #4 can still be associated with a full concentration of solids, even if the permittivity is not uniform. However, from the images in case #5 (even for LBP), it is merely impossible to detect the slug that occurs with a lower concentration, as it can be deduced from Figure 4 or observed on the corresponding camera view in Figure 5. A possible reason is given later on in the next sub-section.

The reconstruction quality improvements from the IBP technique in the cases of slugs #3 and #4 are not convincing due to the simple fact that the sensing area is filled. The gradient of permittivity is sharper than for LBP. It is similar for the settled layer images, but this time, the range of permittivity is "drifted" towards a higher range. Moreover, the difference between no rotation and rotation is more pronounced than for LBP, especially for slug #3. In the case of slug #4, the difference between IBP and LBP with no rotation is more pronounced than for slug #3. We do not have a clear explanation to provide even after inspection of the corresponding data. Despite this problem, it seems clear that the number of iterations and the level of the relaxation factor used for IBP is dependent on many factors that include the type of flow regime (settled layer/slug) and level of noise in the data. In the case of an image with a significant noise level, like in the center of images corresponding to settled layers #1 and #2, a large relaxation factor is needed. However, this seems less crucial in the case of a slug since the sensing area is filled with solids. Nonetheless, a larger number of iterations might be needed to converge towards a more uniform permittivity range. A complementary study is necessary to determine the influence of both parameters on the type of flow regime.

Since the images corresponding to the three different slugs are different, it is worth inspecting the four images that correspond to the set of capacitances used to build the image with the LBP/rotation method. Figure 6 and Figure 7 present these sets of consecutive images for slugs #3 and #4, respectively. The figures use the same permittivity color-bar as in Figure 5.

The last and first images of these figures correspond to the ones shown in Figure 5 for no LBP/no rotation of the corresponding slugs as these sets were acquired during anticlockwise and clockwise rotations, respectively. In the case of slug #3 (Figure 6), one can see that the acquisition of the capacitance set started while a settled layer was present in the sensing area. In contrast, the slug was conveyed within the sensing area during the last set. Therefore, the permittivity gradient is not the result of a real gradient of solids concentration in the slug. Still, the combination of capacitance sets corresponds to a different regime, i.e., settled layer and slug. On the other hand, the sets of capacitance used for slug #4 were acquired with the slug that occurred in the sensing area, as it is shown in Figure 7. One can also see from this figure that three of the four images are very similar (Figure 7b–d), but tilted chronologically in agreement with the clockwise rotation of the sensor.

The qualitative observation was completed with a more quantitative description of the solids concentration. Because of noise presented in the image that generates inaccurate contour of the solids body, especially in the case of the settled layers, a first step concerns the segmentation of the images based on permittivity level, as shown in Figure 8.

For each element of the image with a permittivity of the solid–gas mixture *ε_M_* larger than a level *ε_M,l_*, the corresponding element in the segmented image (or binary image) is set to 1, and the sum of the elements divided by the sensing defines the fractional area *f_A_* of the solids body. Since noise in the image is mostly filtered using IBP, the intersection of the plots with LBP and IBP defines the permittivity level that is used to threshold the images. In the case of settled layers #1 and #2, *ε_M,l_* corresponds to 0.3 (Figure 8a–b), while for slug #4 it corresponds to 0.05. The solids concentration *Φ* is simply calculated after using the following Maxwell–Hewitt law of mixture that considers the solids as the continuous phase and the gas as the dispersed one [32]:(9)$$1-\Phi =\frac{\left(2+\varepsilon_{G}/\varepsilon_{S}\right)\left(1-\varepsilon_{M}/\varepsilon_{S}\right)}{\left(1-\varepsilon_{G}/\varepsilon_{S}\right)\left(2+\varepsilon_{M}/\varepsilon_{S}\right)},$$
where *ε_G_* and *ε_S_* are the normalized permittivity for solids and air, respectively. After sensor calibration, the permittivity of the solid and air are set to 1 and 0, respectively. Since situations occur where the pipe is not entirely filled with solids, Equation (9) is changed to:(10)$$\Phi =f_{A}\left(\frac{3\langle \varepsilon_{M}\rangle }{2+\langle \varepsilon_{M}\rangle }\right),$$
where $\langle \varepsilon_{M}\rangle$ is the mean of normalized permittivity of all elements with a permittivity larger than *ε_M,l_*.

The comparison between the different image reconstruction strategies for the two settled layers #1 and #2 and slug #4 are shown in Table 1.

One can see that the average normalized permittivity, as well as the concentration for the two examples of settled layers, are larger when rotation is taken into account, both for LBP and IBP. Meanwhile, these parameters are also larger for IBP than for LBP. The concentration values for the settled layers are also larger than the corresponding ones estimated from the concentration profile acquired with the first plane of the dual-plane sensor shown in Figure 4 and summarized in Table 1. The same trend is also shown for slug #4 when comparing LBP and IBP using rotation, or LBP with 1 or 4 positions. However, the first image used for IBP presents an unexpectedly large value of the average permittivity, which induces a larger concentration level than expected. This is seen in Figure 5, but no clear conclusion can be drawn, expect the fact that, as it has been mentioned earlier in this section, the relaxation factor and number of iterations may not have been adequate. This possibly leads to a too strong convergence of the rescaled normalized permittivity values towards 1.

There is, therefore, a visible difference between using the rotatable sensor and classical sensor, but it is so far difficult to conclude about a substantial benefit. Static tests on phantoms that simulate the settled layer will be an added value to attest to the improvement of the determination of the concentration level using the rotatable sensor. This work is in progress.

### 4.2. Discussion

The main difference between static and dynamic cases is that the object of interest is in motion. In the case of a “dense-phase” regime in a horizontal channel, this is a mixture of both cases since the settled layer of solids that is present before the net solids transport occurs—the slugs—correspond to a static object. In the case of motion, a parameter that is not so critical in static mode has to be taken into account: the length of the object. Indeed, in the case of a static mode, better reliability of the measurement, in terms of shape retrieval, is achieved when the object is at least as long as the length of the electrodes (the cross-section of the object also must be as uniform as possible along the electrodes). Otherwise, in the dynamic case, the object length should be longer than the electrodes since the acquisition for all combinations of electrodes pairs is generated during a particular time window while the object is moving. If we consider *D* as the length of the electrode (so the minimum length for the sample in static mode), the theoretical extension of the sample to be considered, Δ*D*_th_ is simply a function of the object speed *v* and the frame acquisition rate *f_acq_* of the tomograph, as expressed below:(11)$$\Delta D_{th}=\frac{v}{f_{acq}}$$

A diagram that illustrates this relationship is shown in Figure 9.

In the case of the rotatable sensor (4 frames/s), the extension will be 50 cm and 125 cm for flow speeds of 2.5 ms^−1^ and 5 ms^−1^, respectively. However, for the dual-plane sensor (55 frames/s), these values are considerably lower: around 3.6 cm and 9.1 cm for the corresponding flow speeds specified above. The figure also presents experimental data that were obtained for the three different slugs #3 to #5. The experimental speed *v_exp_* and length extension Δ*D_exp_* were calculated using the cross-correlation data obtained from the dual-plane measurements of the average solids concentration [15,19,33,34,35]. The experimental length extension was estimated by considering the number of successive frames *N*^′^ that present the same speed *v_exp_* as the slug. This can be expressed as:(12)$$\Delta D^{\prime }{\frac{v_{exp}}{f_{acq}{}^{\prime }\frac{d}{N}}}_{exp}$$
where *d* is the distance between two planes of electrodes and *N* is the frame lag estimated from cross-correlation. Values are shown in Table 2.

The time (or frame) lag cannot be established with sufficient precision because of the insufficient temporal resolution of the dual-plane tomographic system. The uncertainty on both *N* and *N′* has been set to one frame to reflect an error in the speed and length extension, as shown in Table 2 and Figure 9. Moreover, the time lag calculated from the cross-correlation is meaningless when the settled layer is present as it does not reflect the reality. One can see from this graph that both slugs #3 and #4 present an extension length, which is greater than the required under the flow speed. It is somehow in agreement with the results shown before. Even for slug #3, the two positions used for the reconstruction with rotation correspond to a settled layer. The positioning of the length extension for slug #5 below the continuous borderline in Figure 9 also explains the problem to recover the correct image of the corresponding slug with the rotatable sensor. This, therefore, reveals one of the main drawbacks of the rotatable sensor for on-line measurement: the constrain of the electrode rotation during measurement acquisition. Of course, this can be improved by using a more powerful stepper motor and, however, can amplify the third problem mentioned below.

A second problem resides in the impossibility to synchronize the beginning of the rotation with the appearance of the slug. It was encountered for slug #3. A possible solution would be to define a communication protocol between the dual-plane ECT and DECaRT which will allow sending a signal to the second acquisition unit to start the sensor rotation when the first acquisition unit detects a slug. However, this will imply considerable work to identify the slug based on capacitance level, but also estimate accurately the flow velocity on-line, so that this solution works.

Despite the authors’ belief that the reconstructed image should be improved from no rotation to rotation scheme, the results presented in this study do not entirely confirm this (even it was shown for the static mode in reference [13]). It is because the recorded capacitance signals are significantly noisier than for a classical sensor. This is mainly linked to the electromagnetic field generated by the stepper motor that strongly alters the capacitance measurements. Indeed, the measurement noise diagnosis shows that DECaRT system SNR is about 50 dB for the classical sensor, but this SNR drops to 45 dB when the stepper motor is connected and rotating. This third problem of the current design is recurrent in using a stepper motor in an electromagnetic environment since the interference can be extremely damaging and very difficult to circumvent. The sensor should, therefore, be improved, especially the shielding, based on electromagnetic compatibility considerations [36].

## 5. Conclusions

This paper presented the first results that deal with the use of a rotatable ECT sensor in dynamic mode during horizontal pneumatic conveying. This feasibility study focused on dense-phase flow, and more precisely on the slug regime. From the results, the following conclusions can be drawn:The use of a rotatable sensor with 12 electrodes and 3 rotation steps achieves relatively satisfying results in terms of better surface delineation and solids concentration for the settled layer, despite the problems related to acquisition speed and noise.So far, the trend seems to show that the concentration level of settled layers and slugs are larger when rotation is used. Static tests on phantoms that simulate settled layers with known concentrations are necessary to attest if the improvement is verified.The current main drawbacks of the rotatable sensor are the rotation speed; also, the electromagnetic field that is generated by the stepper motor, which interferes with the capacitance measurements, is generating a significant drop in the SNR. Improvement of the sensor design is one of the main perspectives of work that is considered for the near future. This will also open the perspective of studying the applicability of the rotatable sensor for other types of flow regime.

## Figures and Tables

**Figure 1 sensors-20-04854-f001:**
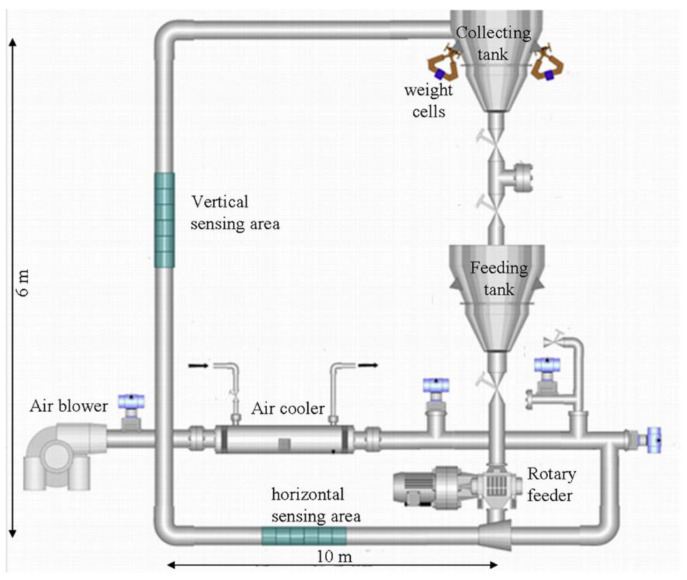
Schematic description of the solid/gas pneumatic flow rig.

**Figure 2 sensors-20-04854-f002:**
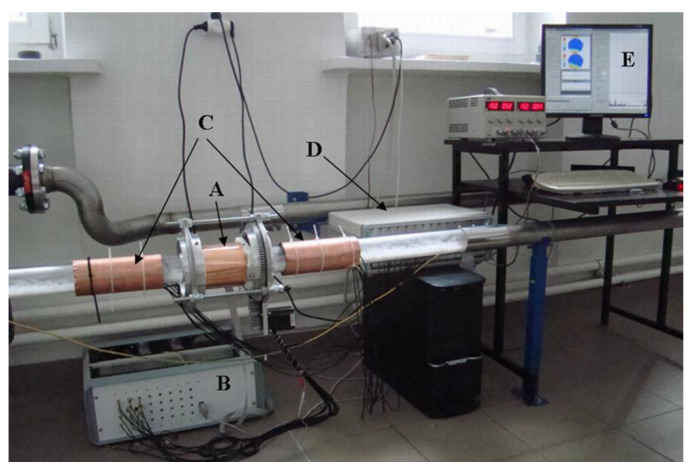
General view of the experimental stand: (**A**) rotatable sensor, (**B**) ECT acquisition unit, (**C**) dual plane for solids flow velocity estimation, (**D**) dual-plane 16 AC-base electrical capacitance tomography (ECT) acquisition unit, (**E**) screen that displays both software of the tomography units.

**Figure 3 sensors-20-04854-f003:**
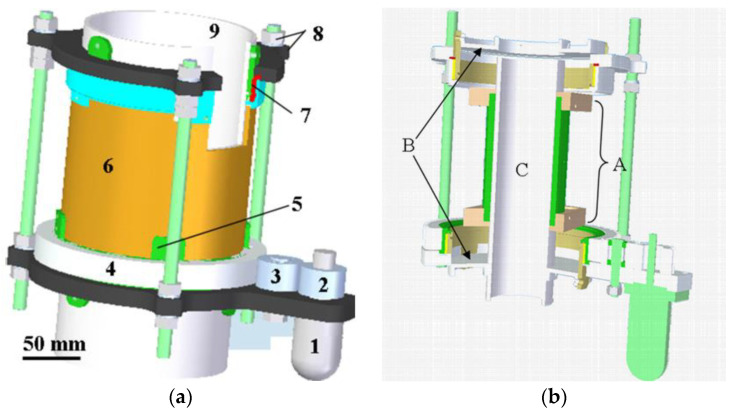
The assembly view of the rotatable sensor: (**a**) 1—stepper motor, 2—driver gear, 3—intermediate gear, 4—transmission gear, 5—supporting mechanism, 6—cylindrical outer tube (electrode holder), 7—sliding bearing, 8—fix mechanism, 9—inner tube (or flow pipe) and the CAD view; (**b**) A—3 parts of the I-shape tube, B—mounting discs, C—flow rig.

**Figure 4 sensors-20-04854-f004:**
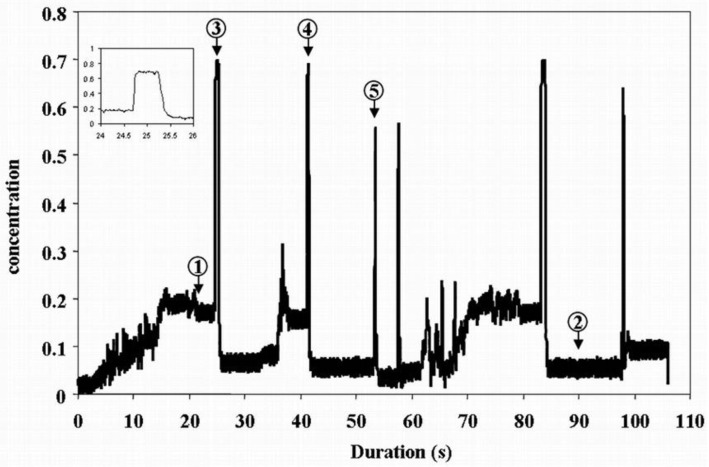
Evolution of the average concentration as a function of time recorded by the dual-plane ECT system. Case of plane 1. Arrows and numbers correspond to the special frame that is also analyzed using the rotatable sensor. 1 and 2 correspond to settled layers while 3–5 to slugs. The concentration profile of slug #3 is also magnified.

**Figure 5 sensors-20-04854-f005:**
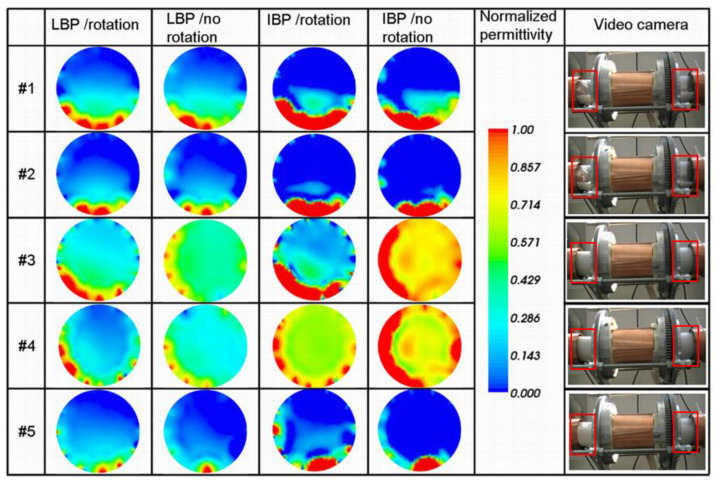
Qualitative comparison of the 2 different image reconstruction techniques (linear back projection (LBP) and iterative back projection (IBP)) taking into account rotation or 1 position for the 5 flow events. The legend corresponding to the normalized permittivity in the different images, as well as screenshots of the corresponding sequence of the recorded movie, are also shown.

**Figure 6 sensors-20-04854-f006:**
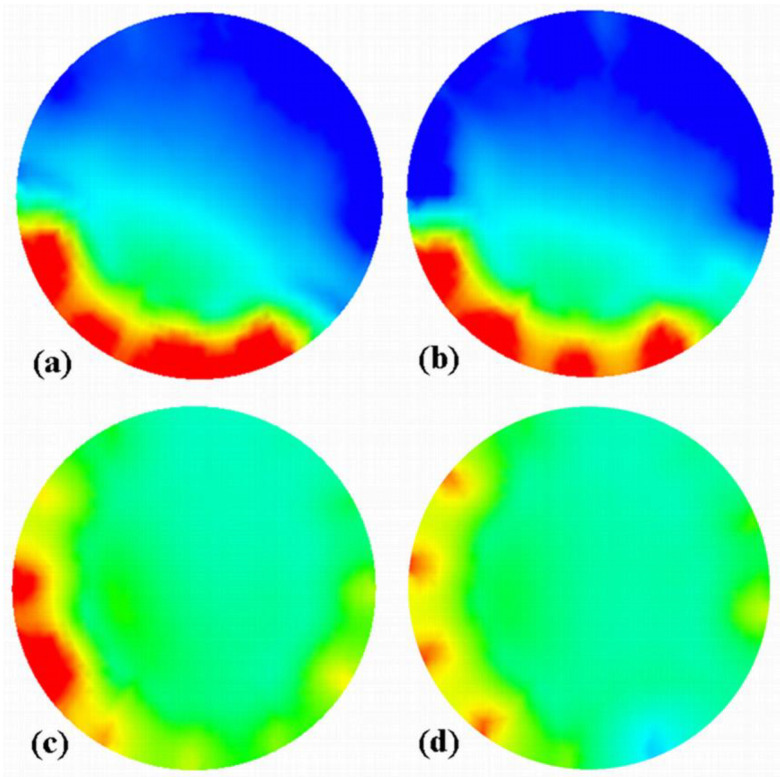
Four images (**a**–**d**) of the slug #3 reconstructed with LBP from capacitance data measured for individual successive positions of the motor (see that the image (**d**) corresponds to the one used for LBP/no rotation shown in Figure 5). This set was acquired anticlockwise.

**Figure 7 sensors-20-04854-f007:**
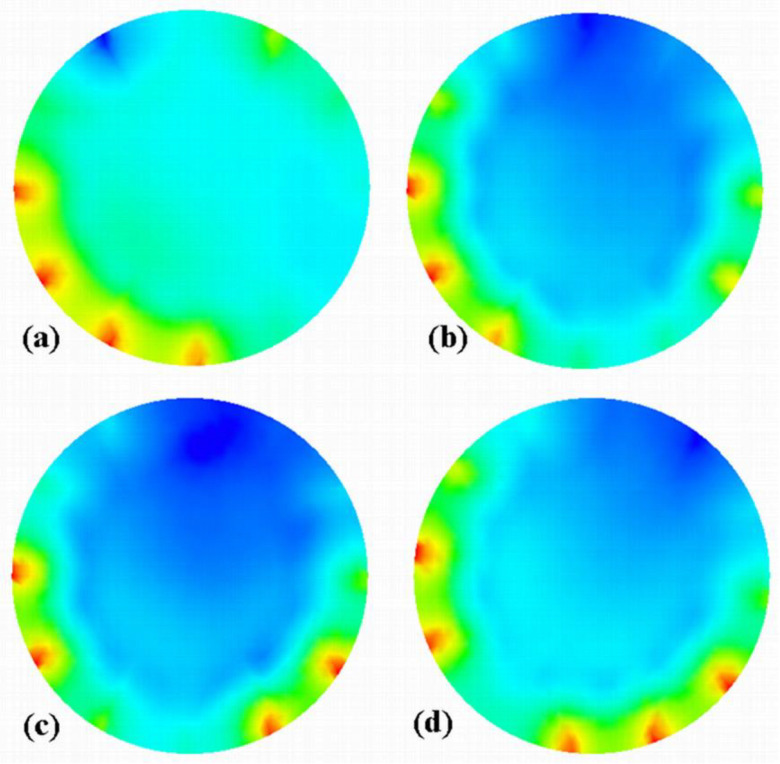
Four images (**a**–**d**) of the slug #4 reconstructed with LBP from capacitance data measured for individual successive positions of the motor (see that the image a corresponds to the one used for LBP/no rotation shown in Figure 5). This set was acquired clockwise.

**Figure 8 sensors-20-04854-f008:**
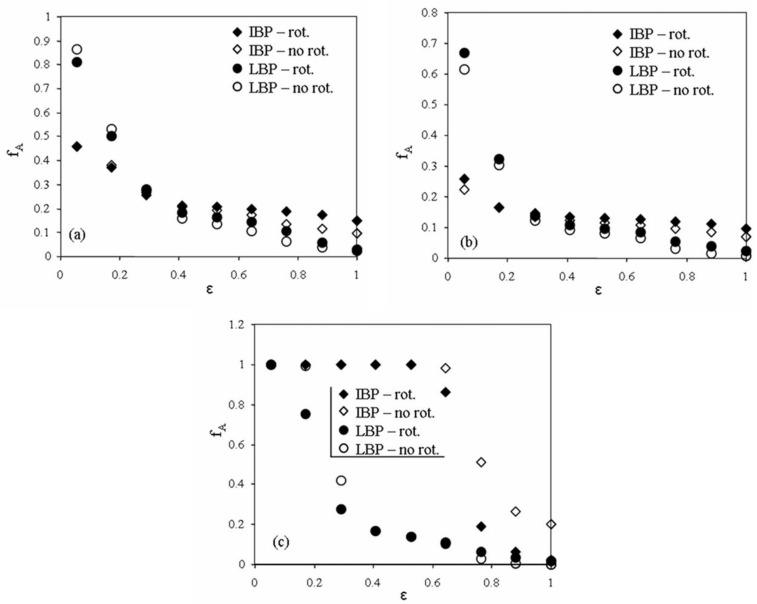
Evolution of the fractional area *f_A_* vs. the normalized permittivity threshold for the 2 image reconstruction methods, taking into account rotation (4 positions) or no rotation (1 position). (**a**) Settled layer #1, (**b**) settled layer #2, (**c**) slug #4.

**Figure 9 sensors-20-04854-f009:**
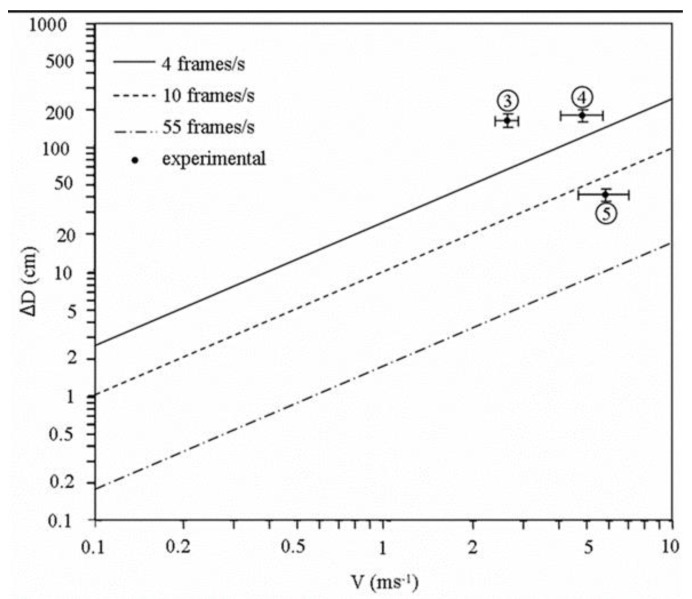
Evolution of the sample length extension ΔD as a function of the solids velocity V for three different acquisition rates. Experimental values corresponding to the ones enumerated in Figure 4 are also shown together with uncertainty bars.

**Table 1 sensors-20-04854-t001:** Comparison between the average normalized permittivity values *<ε_M_>*, fractional areas of solids body *f_A_**,*** and solids concentrations *Φ* for the 4 different image reconstruction methods and settled layers #1 and #2 and slug #4. The concentration *Φ** estimated from the first plane of the dual-plane sensor (see Figure 4) is also shown.

Event	LBP Rotation			LBP No Rotation			IBP Rotation			IBP No Rotation			Estimated Concentration
	$\langle \varepsilon_{M}\rangle$	*fA*	*Φ*	$\langle \varepsilon_{M}\rangle$	*fA*	*Φ*	$\langle \varepsilon_{M}\rangle$	*fA*	*Φ*	$\langle \varepsilon_{M}\rangle$	*fA*	*Φ*	*Φ**
#1	0.63	0.28	0.2	0.55	0.278	0.18	0.828	0.256	0.225	0.706	0.265	0.208	0.177
#2	0.69	0.136	0.105	0.63	0.123	0.089	0.89	0.146	0.135	0.85	0.135	0.121	0.06
#4	0.52	1	0.62	0.466	1	0.566	0.566	1	0.66	0.878	1	0.915	0.69

**Table 2 sensors-20-04854-t002:** Summary of the slug characteristics extracted from cross-correlation in terms of length and speed. Uncertainties are also mentioned.

Event No.	Description	Size of Slug (Frames)	Time Lag (Frames)	Speed (ms^−1^)
#3	Long slug/Moderate speed	35 ± 1	11 ± 1	2.67 ± 0.24
#4	Long slug/Fast speed	21 ± 1	6 ± 1	4.9 ± 0.82
#5	short slug/Fast speed	4 ± 1	5 ± 1	5.88 ± 1.18

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
