# Peer review of "On the Use of a Rotatable ECT Sensor to Investigate Dense Phase Flow: A Feasibility Study"

_sensors, 2020, doi:10.3390/s20174854_

Round 1

Reviewer 1 Report

Dear authors,

the paper entitled "on the use of a rotatable ECT sensor to investigate dense phase flow: a feasibility study" investigates the applicability of a rotating ECT sensor to visualise the dynamics of a two phase flow (gas/particles) in a pipe. The rotation of the sensor increases the number of projections for the reconstruction in order to enhance the quality of the reconstruction. The authors investigate for one experiment the dynamics of the flow and use as reference a video and measurements of a two plane ECT sensor. Two different reconstruction algorithms are compared for the case with and without rotation. It turned out that expected quality enhancement was not achieved and the reasons are discussed in detail.

The paper is clear and concisely written and all difficulties and problems are discussed. Therefore, I recommend to accept the paper after a minor revision, after clarifying following issues:

  1. page 1, line 28: What do you mean by "reduce error scatter"? Here, a reference would be helpful
  2. page 2, line 55: What do you mean by "smallest acquisition and reconstruction times"? Was the reconstruction time reduced? Please make this sentence more clear.
  3. page 3, line 95: In figure 2 only the new sensor is shown, not both types of capacitive sensors, as mentioned in the first sentence. I would swap Figure 3 with Figure 2 and adjust the text accordingly.
  4. Page 3, line 101: What electrode holder was changed? Of the two plane sensor or of the new rotational ECT sensor?
  5. Page 4, figure 2: Please describe more details of figure 2, e.g. the stepper motor. Not all readers have the detailed figures in mind of the article from 2010 (Flow measurement and instrumentation, vol 22, issue 3, pp. 219-227.
  6. page 4, line 131: "The effect ... in Section 0" Please check the reference!
  7. page 5, lines 152-154. Please introduce the abbreviations here, where you introduce for the first time both techniques. To my understanding:
    simple and fast Linear Back Projection (LBP) and slow but accurate non-linear iterative back-projection (IBP).
  8. page 5, lines 166 - 180. It is very difficult to understand, what part is needed for the LBP method and what is "new" for IBP.
    Are equations (4) and (5) used for LBP or are they already only applicable for IBP?
  9. page 6, line 203: could you give an absolute value of the reconstruction time for the experiment?
  10. page 7, line 231: please add the numbers of the event into the text, e.g. " three to slugs (3,4,5) and two to settled (1,2) layers. Please comment also, why you selected these parts.
  11. page 5: The image of the video camera does not really change for the 5 experiments. I suggest to magnify the pipe on the left hand side, where the reader might see the settled and the slug regime, or to remove the images, since they cannot serve as reference for the reconstruction.
  12. page 9: Figures 6 + 7: Please rewrite the caption, it states: figure d is reconstructed with LBP/rotation and also with LBP/no rotation. The reference to figure 5 remains unclear for me. Additionally, I would state that the colorbar is the same as in Figure 5. By the way: why do you show images of the LBP and not of the IBP reconstruction and compare it with the LBP?
  13. page 10, lines 298-300: The description in the text of the Figures does not fit with the caption of the figures.
  14. page 11, lines 344-345: I propose to reconstruct one or two selected images with different relaxation factors and numbers of iterations. Then you may say something about the reconstruction parameters. The information about the convergence of the algorithm would be very beneficial for the contents of the paper, especially, if you have not a very clear improvement of the reconstruction.

Author Response

Dear Reviewer,

We thank you for your constructive comments. We have improved the paper in the light of your feedback. Please find in the attachment the revised version of the paper with highlighted changes we made.

1. page 1, line 28: What do you mean by "reduce error scatter"? Here, a reference would be helpful

We intended to highlight the general trends in the research towards increasing the reconstruction accuracy evaluated with any error (i.e. normalized mean square error ratio or Pearson’s linear correlation coefficient). We used unclear term. Therefore, we thanks to the reviewer for pointing this out. In the article we have corrected this to the relative error and added the reference.

2. page 2, line 55: What do you mean by "smallest acquisition and reconstruction times"? Was the reconstruction time reduced? Please make this sentence more clear.

This sentence was rewritten to clarify dependency between sensor geometry, rotation steps and image accuracy.

3. page 3, line 95: In figure 2 only the new sensor is shown, not both types of capacitive sensors, as mentioned in the first sentence. I would swap Figure 3 with Figure 2 and adjust the text accordingly.

We swapped the figures and corrected their references.

4. Page 3, line 101: What electrode holder was changed? Of the two plane sensor or of the new rotational ECT sensor?

It was done for the rotary sensor. We corrected this in the paper. Regarding the next point of this review, the part “electrode holder” is now mentioned in fig. 3.

5. Page 4, figure 2: Please describe more details of figure 2, e.g. the stepper motor. Not all readers have the detailed figures in mind of the article from 2010 (Flow measurement and instrumentation, vol 22, issue 3, pp. 219-227.

We appended the figure with detailed description

6. page 4, line 131: "The effect ... in Section 0" Please check the reference!

It has been corrected. Should be 4.2.

7. page 5, lines 152-154. Please introduce the abbreviations here, where you introduce for the first time both techniques. To my understanding:
simple and fast Linear Back Projection (LBP) and slow but accurate non-linear iterative back-projection (IBP).

The requested abbreviations have been introduced

8. page 5, lines 166 - 180. It is very difficult to understand, what part is needed for the LBP method and what is "new" for IBP. 
Are equations (4) and (5) used for LBP or are they already only applicable for IBP?

The equations 4 and 5 are defined for LBP algorithm. For proper understanding we added the references to them.

9. page 6, line 203: could you give an absolute value of the reconstruction time for the experiment?

For LBP the reconstruction time for single image is ~15ms. We supplemented this in the article. IPB is around 120 times slower.

10. page 7, line 231: please add the numbers of the event into the text, e.g. " three to slugs (3,4,5) and two to settled (1,2) layers. Please comment also, why you selected these parts.

The numbers have been added to the paragraph. These particular events were chosen for their quasi steady state nature. In that manner, they can be considered as static objects of different kinds. Only in that situation can the effect of the rotatable sensor be tested over the reconstructed image quality. This explanation has been added to the revised version.

11. page 5: The image of the video camera does not really change for the 5 experiments. I suggest to magnify the pipe on the left hand side, where the reader might see the settled and the slug regime, or to remove the images, since they cannot serve as reference for the reconstruction.

We highlighted pipe regions with red rectangles to show the flow changes and the additional explanation has been made.

12. page 9: Figures 6 + 7: Please rewrite the caption, it states: figure d is reconstructed with LBP/rotation and also with LBP/no rotation. The reference to figure 5 remains unclear for me. Additionally, I would state that the colorbar is the same as in Figure 5. By the way: why do you show images of the LBP and not of the IBP reconstruction and compare it with the LBP?

All images (a-d) in figure 6 as well as in figure 7 are reconstructed for measured capacitances obtained only for single step of the motor (single position). Therefore, these images are considered as LBP/no rotation (including image d). Then, these four images (a to d) are combined to have one image for LBP/rotation.

Next, we reference to figure 5 because the image (d) is the same as the LBP/no rotation for #3 and #4 event respectively. However, we added in the text a sentence about the permittivity color-bar which is the same as the one for Figure 5.

We corrected the captions to handle this in a clearer way.

As far as the comparison between LBP and IBP is concerned, we decided not to show the sequence of 4 consecutive images for IBP because the characteristics of the images are similar (e.g. for the event #3 (slug), IBP also reveals the transient change from settled layer to slug. The meaning of these figures was to explain the result of the image reconstruction when the rotation steps are taken into account. In that case, it simply shows that the reconstructed images are “distorted” when 2 objects of different kinds (here settled layer + slug) are used in the reconstruction. The use of IBP does not improve the final image quality.

13. page 10, lines 298-300: The description in the text of the Figures does not fit with the caption of the figures.

Regarding the point 12 of the review we corrected the captions and now the description complements them.

14. page 11, lines 344-345: I propose to reconstruct one or two selected images with different relaxation factors and numbers of iterations. Then you may say something about the reconstruction parameters. The information about the convergence of the algorithm would be very beneficial for the contents of the paper, especially, if you have not a very clear improvement of the reconstruction.

The reviewer's suggestion is accurate. Nevertheless, due to the fact that the article is now too extensive and, in addition, taking into account the different convergence of the algorithm depending on the type of the observed flow, we decided that it would be more legible if we leave the description of the problem for these parameters’ selection, that we made on the basis of the performed experiments. That description can be found in the paragraph (lines 279 - 286; page 8) of the first version of the article (before the review),

Reviewer 2 Report

General comments

The authors investigated the feasibility of a rotatable ECT sensor in a more realistic application that considers dynamic flow. The chosen topic is interesting and worthy of investigation. In addition, the manuscript is well-written and easy to follow. Considering that the current work is an extension of a previous work carried out by the same group on static objects, the current work should be technically sound. My main concern is:

(1) For the LBP algorithm, it seems that the reconstructed images show some artifacts in the near-wall region and the distribution of these artifacts seems to show a certain pattern. I guess the reason should be from the formulation of the LBP algorithm used in this work. As have been confirmed from DOI: 10.1002/aic.15879, an image reconstruction algorithm with and without a division operation can differ much. Therefore, I would like to suggest the authors check on this.

(2) In page 4, how were the minimum rotation time and other parameters calculated? Also. Please explain more clearly on “three rotation steps”.

Author Response

Dear Reviewer,

We thank you for your constructive comments. We have improved the paper in the light of your feedback. Please find in the attachment the revised version of the paper with highlighted changes we made.

(1) For the LBP algorithm, it seems that the reconstructed images show some artifacts in the near-wall region and the distribution of these artifacts seems to show a certain pattern. I guess the reason should be from the formulation of the LBP algorithm used in this work. As have been confirmed from DOI: 10.1002/aic.15879, an image reconstruction algorithm with and without a division operation can differ much. Therefore, I would like to suggest the authors check on this.  

The authors thank the reviewer for pointing this out. In the paper we include the special description that it is possible to reduce the artifacts in the near-walls regions using the division operation as developed in suggested reference. It will be, of course, the aim of our future works.

(2) In page 4, how were the minimum rotation time and other parameters calculated? Also. Please explain more clearly on “three rotation steps”.

The vibrations of the motor have an impact on measurements stability. Using one static phantom we set the angular motor speed decreasing it from 150rpm to 10rpm with a step 10rpm. Then, for each motor speed we collected the set of 100 measurements and performed a statistical analysis. The results indicated the exponential dependency which decreases for the lower values of angular speed. The speed 25rpm was chosen experimentally as a most optimal considering the minimum rotation time and the measurement noise. Because both the motor and the tomography measurement unit were synchronized using author’s software it was possible to measure time required for three rotation steps and four measurements.

Additionally, the SNR analysis can be found in Liu, Z.; Babout, L.; Banasiak, R.; Sankowski, D. Effectiveness of rotatable sensor to improve image accuracy of ECT system. Flow Meas. Instrum. 2010, 21, 219–227 – chapter 5.1.

The “three rotation steps” conception means the following three measurement sensor’s position after rotation excluding the initial position as it has been explained in the text: “Firstly, a set of capacitances is measured. Then, three successive clockwise rotations of 7.5° followed by capacitance measurements are performed”.
